# Deciphering the immune landscape of head and neck squamous cell carcinoma: A single-cell transcriptomic analysis of regulatory T cell responses to PD-1 blockade therapy

Adib Miraki Feriz[1☼], Fatemeh Bahraini[1☼], Arezou Khosrojerdi[2], Setareh Azarkar[1], Seyed Mehdi Sajjadi[3], Edris HosseiniGol[4], Mohammad Amin Honardoost[5], Samira Saghafi[3,6], Nicola Silvestris[7], Patrizia Leone[8], Hossein Safarpour[3]*, Vito Racanelli[9]*

1 Student Research Committee, Birjand University of Medical Sciences (BUMS), Birjand, Iran, 2 Infectious Diseases Research Center, BUMS, Birjand, Iran, 3 Cellular and Molecular Research Center (CMRC), BUMS, Birjand, Iran, 4 Department of Computer Engineering, University of Birjand, Birjand, Iran, 5 Laboratory of Systems Biology and Data Analytics, Genome Institute of Singapore, A*STAR, Singapore, Singapore, 6 Department of Internal Medicine, School of Medicine, BUMS, Birjand, Iran, 7 Medical Oncology Unit, Department of Human Pathology "G. Barresi", University of Messina, Messina, Italy, 8 Department of Biomedical Sciences and Human Oncology, University of Bari "Aldo Moro", Bari, Italy, 9 Centre for Medical Sciences (CISMed), University of Trento and Internal Medicine Division, Santa Chiara Hospital, Provincial Health Care Agency (APSS), Trento, Italy

☼ These authors contributed equally to this work.
* safarpour701@yahoo.com (HS); vito.racanelli@unitn.it (VR)

**Data Availability Statement:** The original data which used in this study, can find online through https://www.ncbi.nlm.nih.gov/geo/query/acc.cgi?

## Abstract

Immunotherapy is changing the Head and Neck Squamous Cell Carcinoma (HNSCC) landscape and improving outcomes for patients with recurrent or metastatic HNSCC. A deeper understanding of the tumor microenvironment (TME) is required in light of the limitations of patients' responses to immunotherapy. Here, we aimed to examine how Nivolumab affects infiltrating $T_{regs}$ in the HNSCC TME. We used single-cell RNA sequencing data from eight tissues isolated from four HNSCC donors before and after Nivolumab treatment. Interestingly, the study found that $T_{reg}$ counts and suppressive activity increased following Nivolumab therapy. We also discovered that changes in the CD44-SSP1 axis, NKG2C/D-HLA-E axis, and KRAS signaling may have contributed to the increase in $T_{reg}$ numbers. Furthermore, our study suggests that decreasing the activity of the KRAS and Notch signaling pathways, and increasing *FOXP3*, *CTLA-4*, *LAG-3*, and *GZMA* expression, may be mechanisms that enhance the killing and suppressive capacity of $T_{regs}$. Additionally, the result of pseudo-temporal analysis of the HNSCC TME indicated that after Nivolumab therapy, the expression of certain inhibitory immune checkpoints including *TIGIT*, *ENTPD1*, and *CD276* and *LY9*, were decreased in $T_{regs}$, while *LAG-3* showed an increased expression level. The study also found that $T_{regs}$ had a dense communication network with cluster two, and that certain ligand-receptor pairs, including SPP1/CD44, HLA-E/KLRC2, HLA-E/KLRK1, ANXA1/FPR3, and CXCL9/FCGR2A, had notable changes after the therapy. These changes in gene expression and cell interactions may have implications for the role of $T_{regs}$ in the TME and in response to Nivolumab therapy.

acc=GSE195832. The code to reproduce the results is available at https://github.com/SafarpourLab/HNSCC.

**Funding:** The authors received no specific funding for this work.

**Competing interests:** The authors have declared that no competing interests exist.

**Abbreviations:** CPI, Anti-PD-1 Checkpoint Inhibitors; HNSCC, Head and Neck Squamous Cell Carcinoma; MSigDB, Molecular Signature Database; scRNA-seq, Single-Cell RNA-sequencing; TAMs, Tumor Associated Macrophages; TCGA, The Cancer Genome Atlas; TME, Tumor Microenvironment; Treg, Regulatory T cells.

## Introduction

Head and neck squamous cell carcinoma (HNSCC) is a highly fatal malignancy arising from the mucosal epithelium of the tongue, mouth, nasopharynx, larynx, and throat, with an annual mortality rate of 40–50% [1, 2]. The disease is associated with significant clinical challenges, including a high incidence of distant metastases (10–30%) and tumor recurrence (30%-50%) [3]. HNSCC pathogenesis is driven by various factors such as tobacco exposure, betel nut consumption, alcohol consumption, consumption of spicy food, dental trauma, sunlight exposure, chronic inflammation, Human Papillomavirus infection, somatic genetic mutations, genetic predisposition, and alterations in the microbiome [1, 4].

Despite substantial technological breakthroughs in HNSCC therapy, the mortality rate remains high [5]. HNSCC management has traditionally relied on surgery, radiation, and systemic chemotherapy, either as monotherapy or in combination [6]. Concurrent chemo/radiotherapy, particularly with cisplatin, is a promising therapeutic option, significantly improving survival outcomes for patients with inoperable tumors [7]. However, the genetic complexity of HNSCC plays a crucial role in dictating patient outcomes, with loco-regional failure representing a significant challenge. The use of cisplatin as a radio-sensitizer has been associated with significant systemic toxicity, limiting its applicability in immunocompromised or frail HNSCC patients [5]. Therefore, there is a growing interest in developing innovative therapeutic strategies that maximize disease control while minimizing treatment-related morbidity.

Over the last decade, the field of cancer immunotherapy has rapidly advanced and has been successfully used to treat various malignancies such as melanoma, breast, colorectal, and lung cancer [8]. The use of immunotherapy in the treatment of HNSCC has also revolutionized the field by improving survival rates and decreasing side effects associated with traditional therapies [8]. With the discovery of immunotherapeutic modalities such as oncolytic viruses, monoclonal antibodies, CAR-T cells, and therapeutic vaccines, the HNSCC treatment landscape is rapidly changing and expanding [9]. These innovative approaches hold tremendous potential for improving patient outcomes and quality of life.

Anti-PD-1 checkpoint inhibitors (CPI) have shown great promise in the treatment of recurrent or metastatic HNSCC [10]. Nivolumab, as a PD-1 CPI, is a commonly used FDA-approved treatment for patients with platinum-restricted recurrent or metastatic HNSCC. However, the success rate of this therapy is still modest, with only around 15% of patients responding to PD-1/PD-L1 inhibitors in previous trials [11]. While most studies on CPI resistance have focused on the immune microenvironment and the deactivation and exhaustion of T and B cells [12], less attention has been given to tumor cell-intrinsic mechanisms of immunotherapy resistance in HNSCC. Thus, to identify drivers of response and resistance to CPI, studying the tumor microenvironment (TME) is crucial.

The TME is a complex and dynamic ecosystem that includes various immune cells that migrate to the tumor site. Among these immune cells, T regulatory cells ($T_{regs}$) play a crucial role in promoting tumor growth by regulating immune system hyper-activation and maintaining tolerance [13]. $T_{regs}$ can suppress anti-cancer responses and promote angiogenesis at the tumor site [14]. However, the prognostic effect of $T_{regs}$ is controversial and varies according to the type of cancer, tumor stage, and treatment [15]. In some cancers, such as melanoma, hepatocellular, cervical, renal, and breast cancer, $FOXP3^+$ $T_{regs}$ infiltration has been associated with decreased overall and disease-free survival [16]. However, the effect of Tregs on HNSCC is still controversial. Some studies have suggested that increased $FOXP3^+$ $T_{regs}$ infiltration leads to improved survival [17–19], while others have reported that it causes decreased survival [20–22]. Therefore, further studies are required to elucidate the role of $T_{regs}$ in HNSCC.

The development of single-cell RNA sequencing (scRNA-seq) has provided a comprehensive examination of the transcriptome profiles of specific cell populations, which has significantly assisted the study of the TME [23, 24]. This approach has also been utilized to investigate the TME of HNSCC, and the results have given researchers a deeper understanding of how cells interact and change at the tumor site [23, 25]. The aim of this study is to investigate the effect of Nivolumab on the function and cellular communication of $T_{regs}$ using scRNA-seq data analysis. By studying the effect of Nivolumab on $T_{regs}$, we hope to gain a better understanding of the mechanisms underlying the response and resistance to anti-PD-1 CPI in HNSCC. This study has the potential to contribute to the development of more effective immunotherapeutic strategies for the treatment of HNSCC.

## Materials and methods

The study overall design and flow process are presented in Fig 1.

### Single-cell RNA sequencing data processing

The scRNA-seq data analysis was performed on four donor tumors from a neoadjuvant study of advanced-stage HNSCC patients who were treated with the anti PD-1 therapy, Nivolumab. The samples were taken before and after the patients received treatment. The primary data of GSE195832 by Obradovic et al., was obtained from the Gene Expression Omnibus (GEO) database (https://www.ncbi.nlm.nih.gov/geo/query/acc.cgi?acc=GSE195832) [26]. This data was based on Illumina NovaSeq 6000 paltform (Homo sapiens).

In the analysis of scRNA-seq data using Scanpy (version 1.9.1), a rigorous pipeline was meticulously executed. Initially, a quality control step identified and eliminated low-quality cells with less than 200 expressed genes and those with more than 20% mitochondrial content. Additionally, genes expressed in fewer than 20 cells were filtered out. Subsequent data preprocessing involved the normalization of raw gene expression counts using the *sc.pp.normalize_-total* function, with a target total sum of 10,000 counts per cell, and a logarithmic

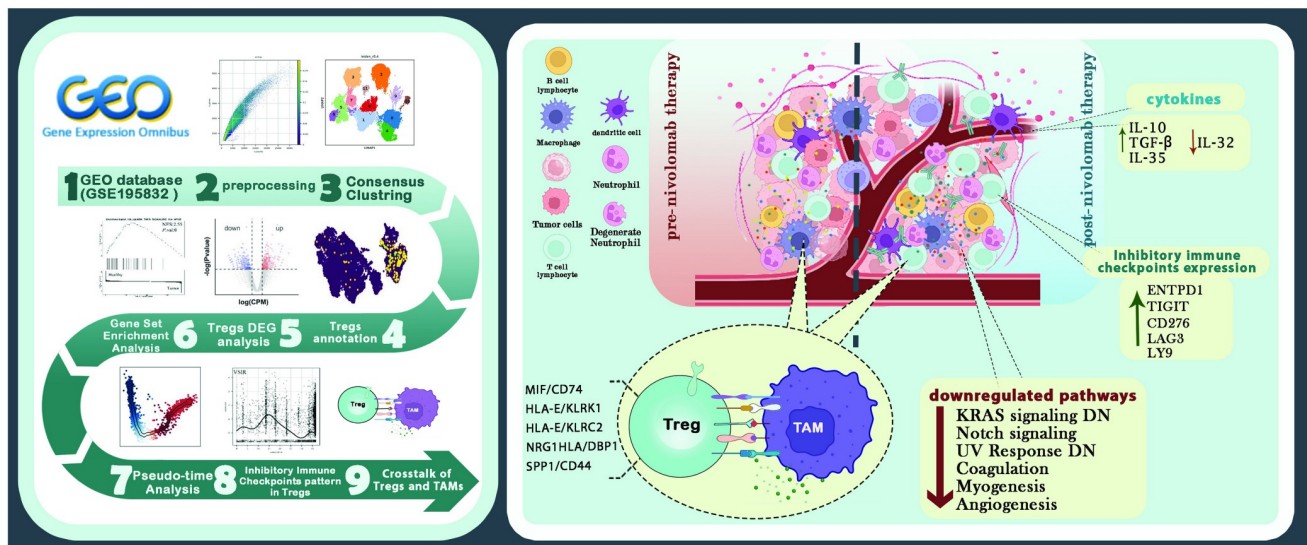

**Fig 1. Schematic visualization of the study.** Figure made by Biorender. Reprinted from [86] under a CC BY license, with permission from Elsevier, original copyright 2023.

transformation to stabilize variance. To capture the most informative genes, Highly Variable Genes (HVGs) were precisely selected using the *sc.pp.highly_variable_genes* function, and retaining the top 4000 genes. These HVGs were then subjected to principal component analysis (PCA) [27] to reduce dimensionality. To address batch effects, the 'combat' algorithm (version 0.3.3) [28] was meticulously applied to harmonize PCA embeddings from eight distinct samples. Utilizing the top 50 principal components, a neighborhood graph was constructed to capture cell-cell similarities (S1 Fig), and the Leiden algorithm was precisely utilized for clustering cells into biologically meaningful groups. Finally, differential expression analysis, leveraging the *sc.tl.rank_genes_group* function with use_raw = True, was conducted to uncover genes with significant expression differences within each cluster.

The regulatory T cells ($T_{regs}$) were annotated with specific markers such as *CD3D*, *FOXP3*, *TIGIT*, and *FANK1* which reported in previous studies [29, 30]. After that, we constructed UMAP embeddings with a minimum distance of 0.5 and a spread of 1.0 to display the most closely similar neighbor graph [31].

### Differentially expressed genes and cell cycle analysis of $T_{regs}$

DEGs were evaluated using the *sc.tl.rank_genes_groups* function using paired t-test method to discover differences between $T_{regs}$ in untreated and treated samples. For downstream analysis, genes having an adjusted *p*-value of $< 0.05$ and fold change $>|1|$ were chosen. The *p*-value was adjusted using Benjamini-Hochberg.

In order to predict the cell cycle stage, the S and G2M-specific genes were scored using the Scanpy function (*scanpy.tl.score_genes_cell_cycle*). The S- and G2M label for each individual cell are determined by the class with the highest score. If neither the S-score nor the G2M-score exceeds 0.5, the cells are said to be in the G1 phase. The reference genes for the cell cycle phase that are utilized for scoring are included in the Kowalczyk et al. study [32].

### Enrichment analysis of DEGs of $T_{regs}$ subpopulation

In our study, we performed gene enrichment analysis using two distinct methods to comprehensively evaluate the biological implications of DEGs within the Tregs subpopulation. First, we employed Over-Representation Analysis (ORA) with Enrichr (https://maayanlab.cloud/Enrichr/), which allowed us to assess whether the DEGs were significantly associated with specific Gene Ontology (GO) terms of biological processes. This approach highlighted over-represented functional categories among the DEGs. Additionally, we utilized Gene Set Enrichment Analysis (GSEA) with WebGestalt (http://www.webgestalt.org/option.php), a robust tool that assessed whether our DEGs exhibited coordinated and statistically significant expression patterns within predefined gene sets from the MSigDB. This dual approach ensured a comprehensive exploration of the functional relevance of the DEGs, providing valuable insights into the underlying biological processes and pathways within the Tregs subpopulation.

### Pseudotemporal ordering of single cells

We conducted pseudotime analysis using scFates v0.8.1 (https://pypi.org/project/scFates/), an analytical tool seamlessly integrated with Scanpy and notable for its GPU-accelerated capabilities, facilitating faster and more scalable inference. Pseudotime analysis involves the estimation of cellular progression along developmental trajectories and examination of gene expression pattern across pseudotime, and scFates is well-suited for this purpose. It allowed us to infer pseudotime values for individual cells, providing insights into their developmental states. By comparing these pseudotime genes with DEGs, we identified Treg-specific genes within the

trajectory. This approach helped unravel the temporal dynamics and critical cellular transitions within the biological system under investigation.

## Expression pattern of inhibitory ICs in $T_{regs}$

To find the expression behavior of ICs in $T_{regs}$, we used DEG analysis to compare the expression patterns of a broad panel of inhibitory ICs, including *TIGIT*, *LY9*, *PDCD1*, *LAG3*, *CTLA4*, *CD276*, *NT5E*, *PDCD1LG2*, *CD274*, *IDO1*, *VSIR*, *HAVCR2*, and *ENTPD*, in untreated and treated samples. The cluster specificity of their expression was then assessed using UMAP, and IC expression was then displayed using pseudo-time. Then, we used the GEPIA database, a user-friendly web-based tool designed for the analysis and visualization of gene expression pattern across multiple cancers from TCGA and the Genotype-Tissue Expression (GTEx) projects, to determine if the expression patterns of the pertinent ICs matched those of their counterparts in the TCGA PCa dataset.

## Cell-cell communication analysis

Cell-cell interaction was investigated using SquidPy [33], which provides analytical techniques for depositing, modifying, and interactively clarifying single-cell RNA sequencing data. It employs a productive re-implementation of the CellPhoneDB technique [34]. CellPhoneDB is particularly notable for its ability to handle a substantial number of interacting cell pairs, often exceeding 100,000, and it accommodates the analysis of interactions across diverse cluster combinations, frequently numbering over 100 clusters. To ensure the reliability of our findings, we rigorously considered interactions where both ligand and receptor genes were expressed in at least 10% of the cells within our scRNA-seq dataset.

# Results

## HNSCC TME cell fractions

We reanalyzed a published scRNA-seq dataset from eight HNSCC tissues in order to comprehend the heterogeneity in the patient response to CPI therapy. After removing cells that failed Quality Control (QC), a total of 53,730 cells remained for downstream analysis (pre-Nivolumab therapy: n = 27223, post-Nivolumab therapy: n = 26507) (Fig 2A and 2B). Cells were represented as twelve different clusters using uniform manifold approximation and projection (UMAP) and unsupervised graph-based clustering (Fig 2C).

Based on the expression of canonical gene markers including: *CD3D*, *FOXP3*, *TIGIT*, and *FANK1*, cluster four was identified as the $T_{reg}$ population (Fig 2D). Interestingly, examination of the The Cancer Genome Atlas (TCGA) cohort indicated that *FOXP3* and *TIGIT* were significantly up-regulated in HNSCC samples compared to healthy (Fig 2E).

We found that tumor cells showed a notable increase in $T_{reg}$ cluster compared to pretreatment tumors (Fig 2F and 2G).

## DEG analysis of $T_{regs}$

The gene expression level in $T_{regs}$ before and after treatment with Nivolumab was determined by differentially expressed gene (DEG) analysis (Fig 3A). Pseudogenes, mitochondrial encoded genes, and ribosomal genes were eliminated, and 4089 Differentially expressed genes (DEGs) (adjusted p-value <0.05), including 284 up-regulated genes and 3805 down-regulated genes, were identified (Fig 3A).

Based on previous studies [35], heterogeneity within $T_{reg}$ cells has been characterized by a bimodal distribution of *TNFRSF9*, a known marker for $T_{reg}$ activation. To investigate whether

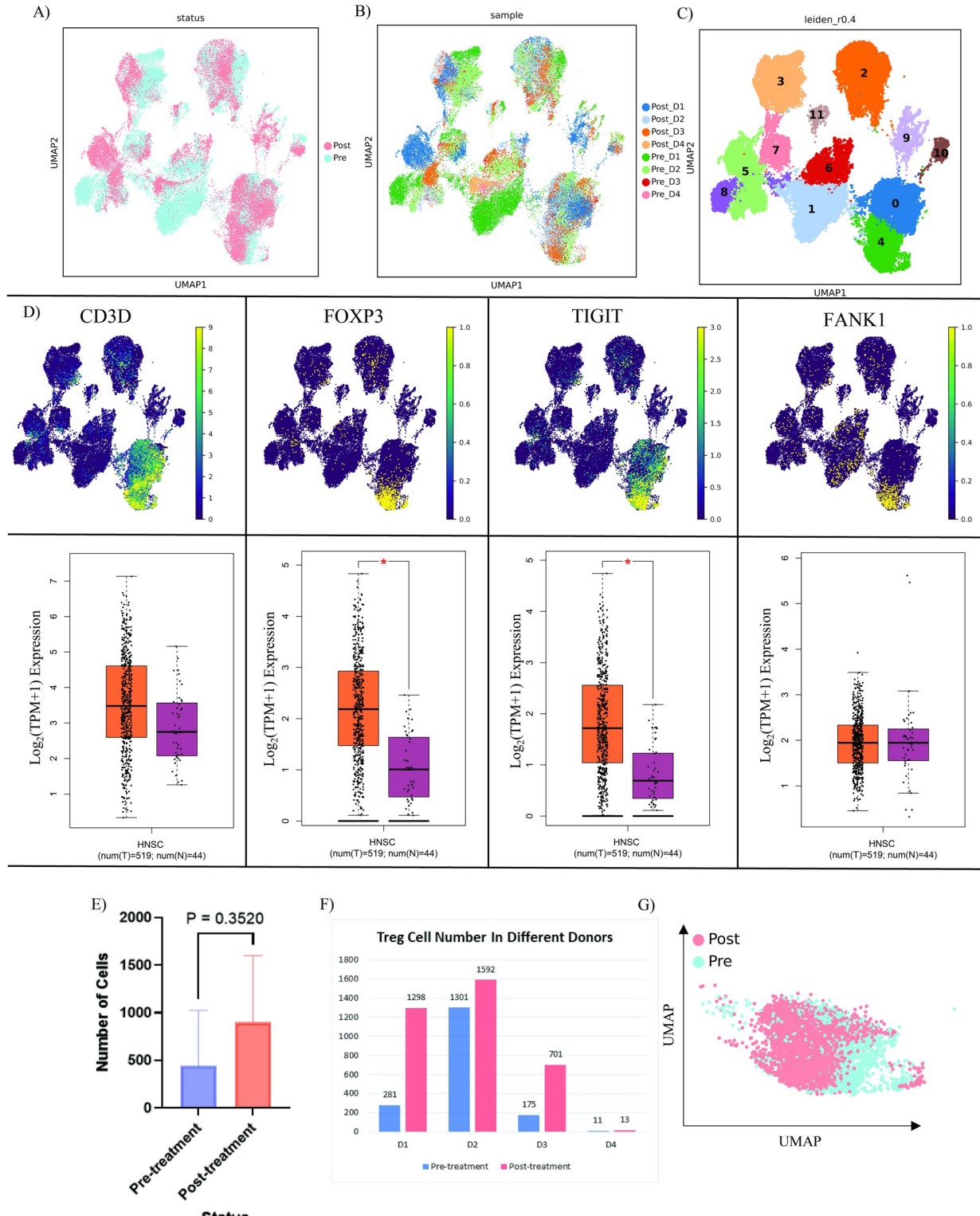

**Fig 2. Characterization of T_regs in HNSCC TME in pre and post Nivolumab therapy. A)** UMAP visualization of cells in two statuses; **B)** UMAP visualization of cells according to their originated samples; **C)** UMAP visualization of HNSCC TME clusters based on Leiden clustering; **D)** UMAP visualization of T_regs markers and related expression pattern of them in TCGA tumor bulk dataset; The box plots display the median mRNA expression levels of Treg markers in HNSCC tumors (orange) compared to normal tissues (red). The values on the axes are presented in Log2 (TPM+1), where TPM (Transcripts Per Million) quantifies gene expression while considering transcript length and sequencing depth. Log2

transformation makes the data more interpretable and robust for visual comparisons. The "+1" avoids issues with zero TPM values, ensuring all values are positive. The * red indicates a p-value less than 0.01, implying statistical significance. The abbreviations T and N denote tumor and normal tissues, respectively. **E)** Histogram representation of $T_{regs}$ cell count in pre (blue) and post (orange) Nivolumab therapy; paired t-test with p-value < 0.05, Error bars indicate standard deviation; **F)** Histogram representation of $T_{regs}$ cell number in different donors in pre (blue) and post (orange) Nivolumab therapy; **G)** UMAP visualization of $T_{regs}$ in two different statues.

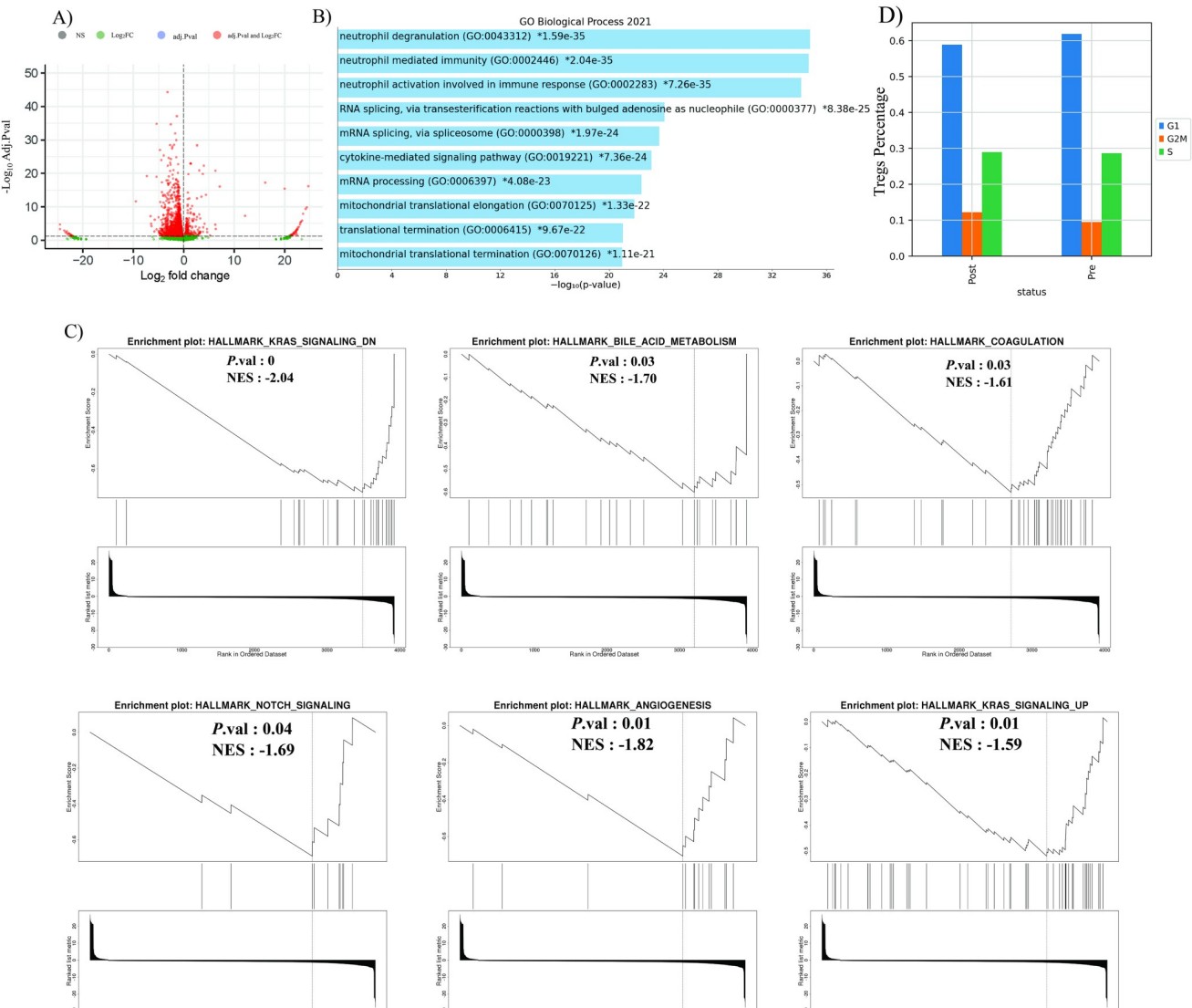

**Fig 3. DEGs, enrichment, and cell cycle analysis of $T_{regs}$ in HNSCC TME. A)** Volcano plot of the Treg's DEGs after Nivolumab therapy. Genes regulated with a fold change >|1| and Adj. p.value < 0.05 are highlighted in red showing the indicated fold changes derived from t-test statistic; **B)** GO Biological Processes of Treg's DEGs using ORA. The figure contains bar charts showing the results of the enrichment analysis of GO developed using Enrichr. For each term, the x-axis indicates the -log10 (p.value); **C)** GSEA analysis of $T_{regs}$ DEGs. GSEA analysis using hallmark gene sets from the molecular signature database for the transcriptional difference between pre and post Nivolumab therapy. NES = normalized enrichment score; **D)** Bar plot of the percentage of $T_{regs}$ in different phases (G1: Growth 1 phase, S: DNA Synthesis phase, G2M: Checkpoint, Mitosis phase) and statues.

this heterogeneity is associated with response to treatment, we further performed DEG analysis of TME cells. Compared to pre-treatment, we observed that tumor cells in TME of post-treatment samples consistently expressed *TNFRSF9* at higher level. Also, the result of DEG analysis of $T_{reg}$ cells between two statuses indicated that genes associated with immunosuppressive functions including *TNFRSF4*, *ENTPD1*, *REL* and *LAYN* were downregulated in post treatment $T_{regs}$. Notably, we discovered numerous immediate early genes among $T_{reg}$ DEGs, including *NR4A2*, *DUSP1*, *FOSB*, *FOS*, *JUN*, and *JUNB*. These genes are quickly activated in response to stimuli, with no or little nascent protein synthesis [36].

Biological Process (BP) analysis of $T_{regs}$ in post treatment centered around the neutrophil degranulation, neutrophil immunity, neutrophil activation, and the cytokine-mediated signaling pathway (Fig 3B). Furthermore, (Molecular Signatures Database) MSigDB analysis showed the negative regulation of KRAS signaling, Notch signaling, bile acid metabolism, coagulation, and angiogenesis in $T_{regs}$ after Nivolumab therapy (Fig 3C).

We found no significant increase in the Treg cluster's S.Score (module score of genes associated with the S phase of the cell cycle) or G2M.Score (module score of genes associated with the G2M phase of the cell cycle) after treatment with Nivolumab, indicating that this unique transcriptional profile was not due to active cell cycling (Fig 3D).

## Pseudo-time trajectory analysis of $T_{reg}$ dynamic changes

Given the heterogeneity of the TME, we performed a pseudo-temporal reconstruction using scFates to determine the lineage structures and pseudo-temporal variables of the HNSCC TME. There were eight unique nodes found based on transcriptional alterations in pseudo-time trajectory analysis, which node number five mapped on $T_{reg}$ cluster. The top 10 up-regulated genes associated with this node included *DUSP4*, *TRBC2*, *CD7*, *GZMA*, *CD2*, and *TRAC*. Also, *CCL5* as an important chemokine was among these specific genes that indicating stimulation of $T_{reg}$ cells presumably by interferons (Fig 4A). Furthermore, enhanced production of CD3E/D, may reveal epigenetic change in the $T_{reg}$ cells. *IL-32* was another gene with strong expression in the $T_{reg}$ cluster. Thanks to mRNA alternative splicing, this cytokine has nine distinct isoforms and is now recognized as a key pro-inflammatory factor.

When the list of top 10 up-regulated genes associated to $T_{regs}$ node were compared with the $T_{reg}$ DEG list between two statuses, we found that four genes including *GZMA*, *CD2*, *IL-32*, and *TRAC* were also dys-regulated significantly in $T_{regs}$ before and after treatment (Fig 4B). In this regard, post-treatment $T_{regs}$ demonstrated a markedly decrease in *IL-32*, *CD2*, and *TRAC* expression while *GZMA* was up-regulated in response to Nivolumab (Fig 4B).

To determine if these alteration in expression of *GZMA*, *CD2*, *IL-32*, and *TRAC* is a distinct feature of $T_{reg}$ cells, we further assessed the expression of these four genes in all clusters across the pseudo-time. Although these genes were expressed slightly in certain clusters, the $T_{reg}$ cluster exhibited higher levels of expression than the other clusters (Fig 4C).

## IIC expression pattern in $T_{regs}$

Considering the importance of IICs in suppressing immune responses, we compared the expression of 14 common IICs in $T_{regs}$ before and after Nivolumab therapy (Fig 5A). Among the IICs examined, only five were significantly altered between two statuses (Fig 5B). *TIGIT*, *ENTPD1*, and *CD276* and *LY9* levels were significantly lower in $T_{regs}$ after therapy, while *LAG-3* showed an increased expression level (Fig 5C). However, the expression of *PDCD1* (*PD-1*), as the target of Nivolumab, did not significantly dys-regulate before and after the treatment (Fig 5C).

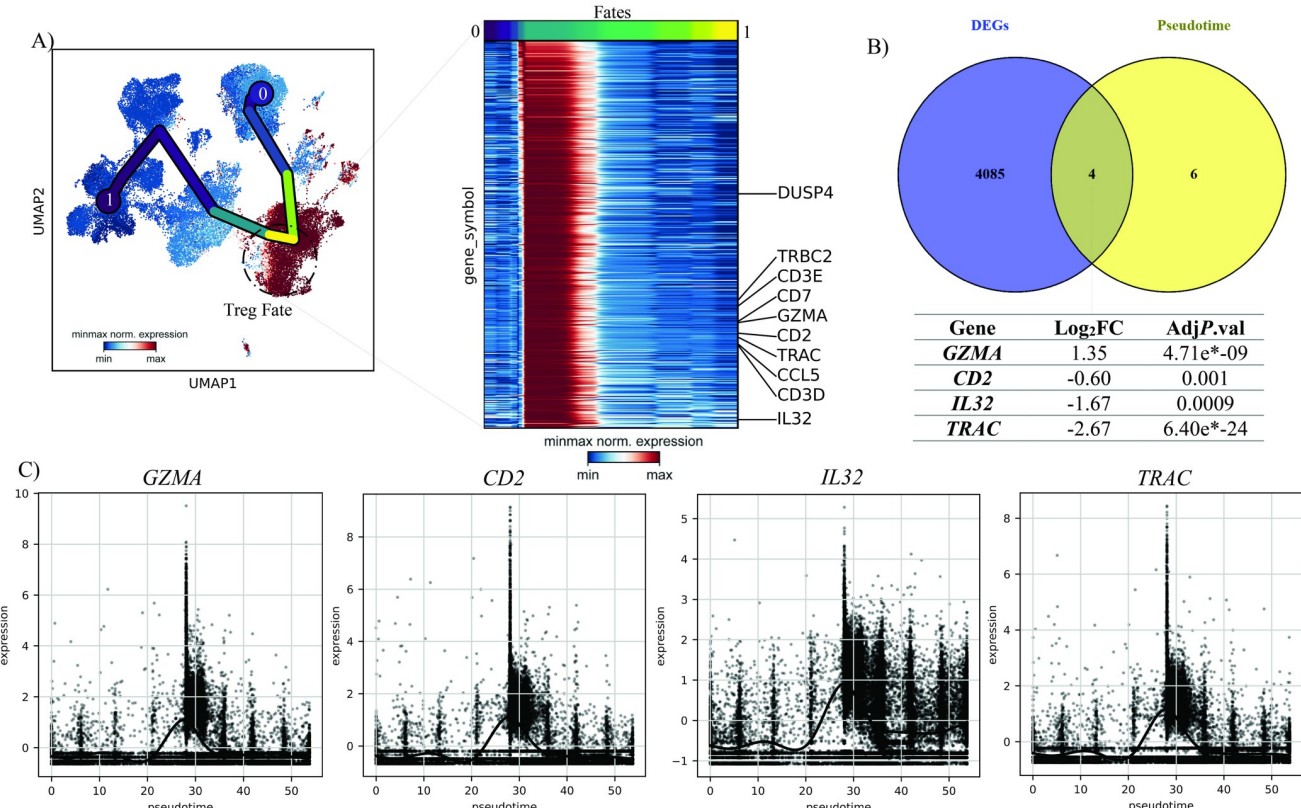

**Fig 4. Pseudo-temporal analysis of HNSCC TME. A)** Top ten up-regulated genes of T$_{regs}$ during the pseudo-time; UMAP presentation illustrating the dynamics of the HNSCC TME and its transition states as they progress through pseudo-time. The lineage tracking initiates at the 0 branch and concludes at the 1 branch. The heatmap depicts genes that experience upregulation within the Treg branch as pseudo-time advances. The highlighted genes comprise the top ten upregulated genes specific to the Treg branch compared to other branches during pseudo-time analysis. **B)** Venny plot of the top ten up-regulated genes of T$_{regs}$ and T$_{regs}$ DEGs; **C)** The expression pattern of four similar genes between the top ten up-regulated genes of T$_{regs}$ and T$_{regs}$ DEGs along the pseudo-time.

UMAP analysis indicated that among the five mentioned IICs, *TIGIT* and *LAG-3* have a higher expression level in T$_{regs}$ than other clusters (Fig 5D). Nevertheless, according to the UMAP results, *LAG-3* expression has increased and *TIGIT* expression has decreased in T$_{regs}$ following treatment with Nivolumab. Based on TCGA HNSCC tumor bulk dataset, patients with HNSCC had higher *TIGIT*, *LAG3*, and *CD276* expression levels than healthy individuals (Fig 5E).

## The cell-cell interaction of T$_{regs}$

We next identified ligand-receptor pairs between other cells and T$_{regs}$ before and after Nivolumab treatment using Squidpy algorithm. This python-based tool includes a database of ligand-receptor interactions as well as a statistical model for identifying relationships between two cell types that are enriched from single-cell transcriptomics data. Results of the analysis demonstrated a dense communication network among cluster two and T$_{reg}$ cells both before and after Nivolumab therapy (Fig 6A). The *MARCO*, *CD14*, *FCGR3A*, and *CD163* genes, which are unique to tumor-associated macrophages (TAMs), were expressed in cluster two (Fig 6B). Among the interactions, five pairs *SPP1/CD44*, *HLA-E/KLRC2*, *HLA-E/KLRK1*, *ANXA1/FPR3*, and *CXCL9/FCGR2A* had shown notable changes after the therapy (Fig 6C). We

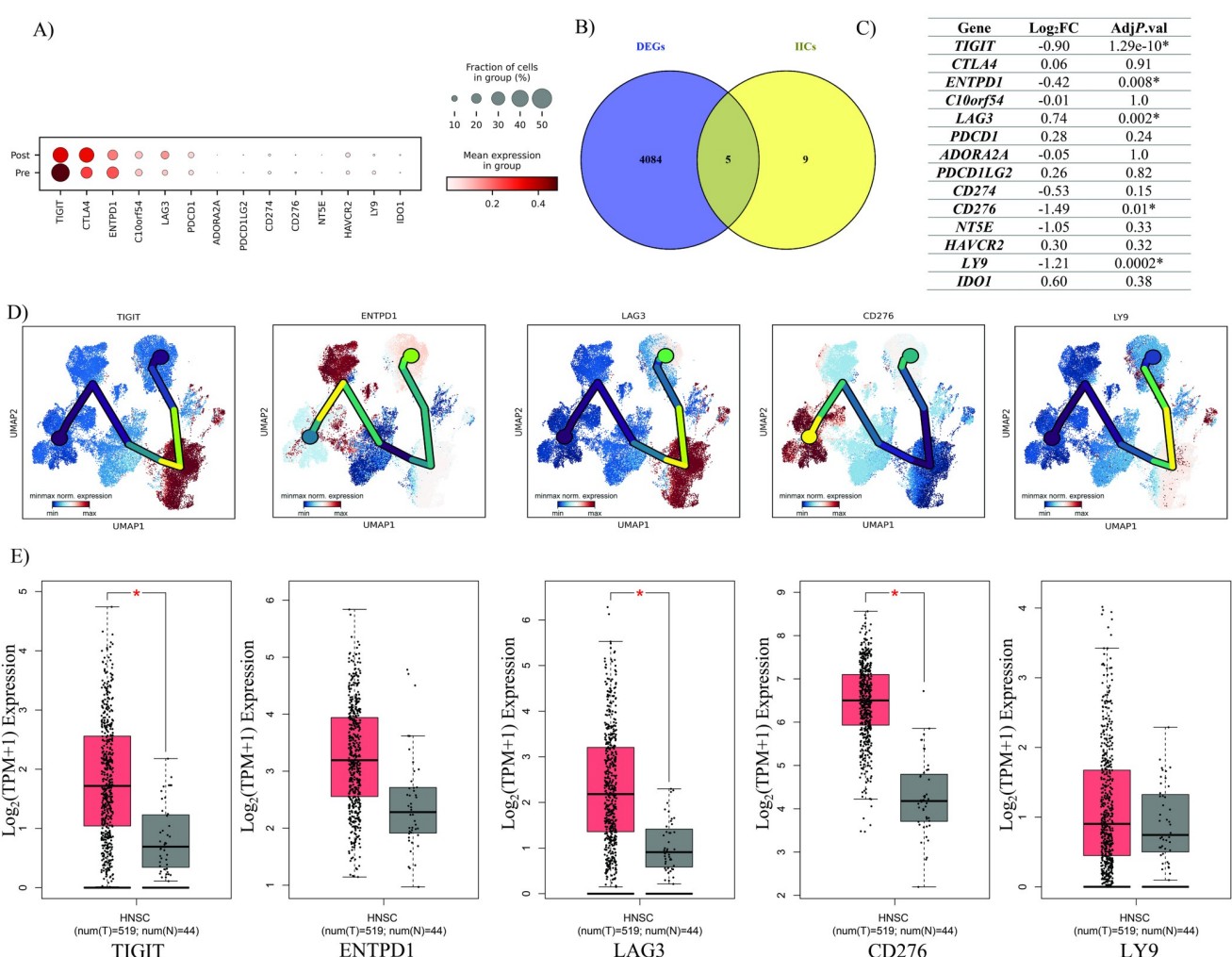

**Fig 5. Characterization of expression pattern of IICs. A)** Dot plot of the expression pattern of IICs in T$_{regs}$ before and after Nivolumab therapy; **B)** Venny plot of the significant IICs in T$_{regs}$; **C)** The log$_2$FC and Adj P.val of IICs in T$_{regs}$ between two statuses. **D)** The expression pattern of significant IICs along the pseudo-time; Yellow branches shows highest expression of the gene across the pseudo-time. **E)** The expression pattern of significant IICs in TCGA tumor bulk dataset, The box plots display the median mRNA expression levels of T$_{reg}$ markers in HNSCC tumors (represented by the orange plots) and the corresponding normal tissues (represented by the red plot). The values on the axes are represented in units of Log$_2$ (TPM+1). The * red indicates a p-value less than 0.01, implying statistical significance. The abbreviations T and N denote tumor and normal tissues, respectively.

identified hallmark associations for these ligands and receptors using MSigDB, focusing on the top 5 pairs (10 genes). The analysis was conducted using ORA and revealed connections to various cellular pathways, including those related to inflammation response, apoptosis, glycolysis, angiogenesis, and signaling pathways such as IL-6/JAK/STAT3 signaling, IL2/STAT5 signaling, and TNFα signaling via NF-κB (Fig 6D).

## Discussion

Not all T cells participate in the development of anti-cancer responses. The TME's development is significantly influenced by a specific subset of T cells that express *CD4*, *CD25*, and *FOXP3* markers and is known as the "T$_{reg}$" subset [37]. These cells, which in normal circumstances assist in preventing the development of chronic inflammation and autoimmune

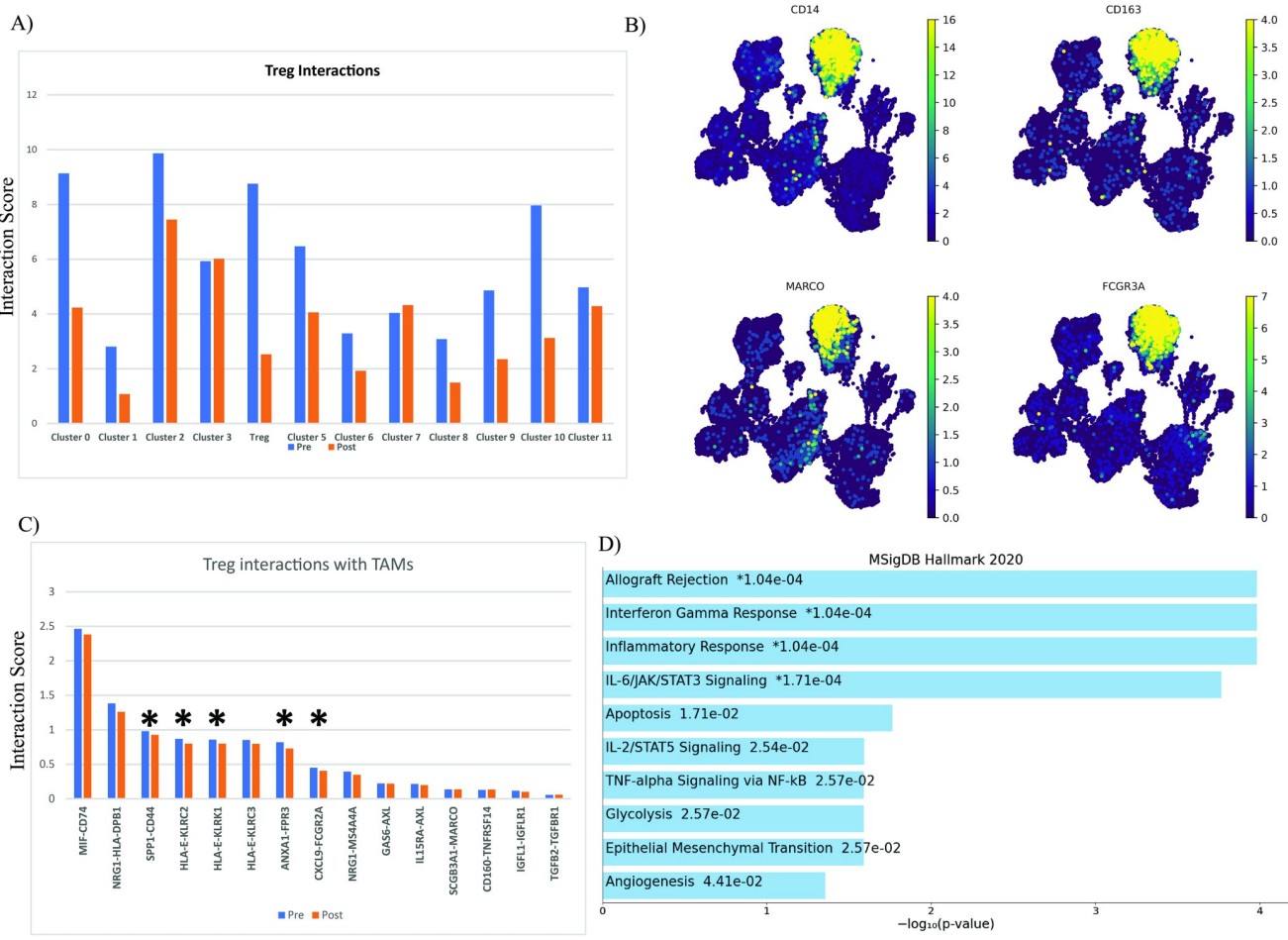

**Fig 6. Cell-cell interaction of T$_{regs}$ in HNSCC TME. A)** Bar plot of the interaction scores of T$_{regs}$ in Pre and Post treatment. Interaction score: sum of all expression of all ligand and receptors which were involved between Treg and others; **B)** UMAP visualization of TAMs markers; **C)** Ligand/Receptors which are significant between T$_{regs}$ and TAMs. Interaction score: Top highly expressed interactions between Treg and TAMs; **D)** MSigDB analysis of the ligand/receptors pathways. The figure contains bar charts showing the results of the enrichment analysis of MSigDB developed using Enrichr. For each term, the x-axis indicates the -log10 (p.value).

diseases, promote the growth and development of tumor cells [38]. T$_{regs}$, in addition to inhibiting CD4$^+$ and CD8$^+$ T cells from becoming activated, enhance angiogenesis to improve the delivery of oxygen and nutrients to the tumor site [39]. As a result, numerous studies have indicated that targeting T$_{reg}$ immunosuppressive mechanisms or eliminating these cells is an essential cancer treatment target [40–42].

T$_{regs}$ infiltrating TME, on the other hand, had inconclusive results in some cancers, such as HNSCC [43]. Some studies have shown the poor prognosis of HNSCC patients with a high FOXP3$^+$ T$_{reg}$ cell infiltration [20, 21]. In contrast, others have highlighted the correlation between the high recruitment of these cells and better overall survival and local control [18, 44].

In this study, we re-analyzed a scRNA-seq data of HNSCC patients treated with Nivolumab to examine the cellular and molecular dynamics of T$_{regs}$ in the TME.

Previous studies have already established several markers for T$_{reg}$ cells such as *CD3D*, *FOXP3*, *FANK1*, and *TIGIT* using single cell sequencing [29, 45], which we also employed in

our study to accurately validate T$_{reg}$ cell annotation. In addition, we discovered that *FOXP3* and *TIGIT* are expressed at higher levels in HNSCC patients by analyzing the TCGA tumor bulk dataset (Fig 2D). TIGIT$^+$ T$_{regs}$ are highly immunosuppressive, persistent, and concentrated in cancers, according to Julien et al [46]. Additionally, FOXP3$^+$ T$_{regs}$ are capable of inhibiting the proliferation of autologous CD4$^+$ CD25$^-$ T cells [47].

Our results demonstrate that both collectively and individually, T$_{reg}$ numbers increased following treatment with Nivolumab (Fig 2E and 2F). According to a comparison of the cell cycle of T$_{regs}$ before and after treatment, evidence showed that T$_{reg}$ proliferation increased dramatically following Nivolumab treatments (Fig 3D). The increase of T$_{regs}$ after treatment with anti-PD-1 has also been confirmed in the study of Kamada et al [48]. In addition, Xiong et al. showed that Nivolumab treatment increases the number of FOXP3$^+$ CD4$^+$ T-cells in peripheral blood [49].

There are some crumbs of evidence to support the cross-talk between neutrophils and T$_{regs}$. According to studies, T$_{regs}$ facilitate neutrophil migration into the TME by generating *CXCL8* [50]. Furthermore, T$_{reg}$-secreted *IL-10* and *TGF-β* can induce neutrophil polarization and conversion to tumor-associated neutrophils (TANs) [51]. T$_{reg}$ also suppress the phagocytosis ability of neutrophils [52]. Following the discovery by Eruslanov et al. that TANs are not immunosuppressive in the early stages of cancer but rather stimulate T cell responses, the significance of neutrophil-T$_{reg}$ cross-talk in TME doubles [53]. Our results showed that the mechanisms of T$_{reg}$ involved in the regulation of neutrophil activity, such as degranulation and mediating immunity, had changed after Nivolumab therapy (Fig 3B).

According to GSEA analysis, KRAS signaling, Notch signaling, and angiogenesis activity were all reduced in T$_{regs}$ following Nivolumab treatment (Fig 3C). Abundant molecules in the KRAS signaling pathway participate in intracellular signaling and regulate cell differentiation, proliferation, growth, and apoptosis [54]. Uncontrolled proliferation and an elevated risk of malignancy can result from mutations in any of the genes responsible for producing these molecules [55]. Numerous studies have also shown that RAS signaling pathway alterations impact T cell function [56–58]. Mor et al. found that inhibiting Ras increases the level of *FOXP3* in pre-existing T$_{regs}$ and improves the conversion of CD4$^+$ CD25$^-$ T cells to CD4$^+$ CD25$^+$ FOXP3$^+$ T cells [59]. Additionally, this work demonstrated that RAS inhibition improves T$_{regs}$' immunosuppressive function in a *FOXP3*-dependent manner, which allowed RAS-deficient T$_{regs}$ to prevent the development of diabetes in a mouse model of the disease by up to 70% [59]. Our research showed that Nivolumab therapy reduced KRAS signaling in T$_{regs}$, which could cause an increase in the expression of *FOXP3* and enhance the immunosuppressive ability of T$_{regs}$.

The Notch signaling pathway is one of the most evolutionarily conserved pathways and is activated only by cell-to-cell communication [60]. Rong et al. demonstrated that the suppressive activity of T$_{regs}$ is adversely affected by the activation of the Notch signaling pathway [61]. Activation of the Notch signaling pathway, in other words, causes T$_{regs}$ to express less *TGF-β*, *IL-27b* (a component of *IL-35*), and *PD-1* [61]. T$_{regs}$ treated with Notch ligands produced more *IFN-γ* and were less capable of inhibiting T-effector proliferation and preventing the release of *TNF-α* and *IFN-γ* [61]. Our results demonstrated that Nivolumab treatment decreased the Notch signaling pathway in T$_{regs}$, which may have increased their immunosuppressive ability.

Accurate study of diverse gene expression in scRNA-seq utilizing pseudotime can aid in better understanding the specificity of any therapy, leading to the identification of novel genes as critical player targets. Accordingly, we found that the expression levels of four genes *GZMA*, *CD2*, *IL32*, and *TRAC* were altered in T$_{regs}$ after Nivolumab treatment (Fig 4B). The expression level of *GZMA* increased while the expression of other genes decreased. It has long been

known that tumor and virus-infected cells can be killed by (Natural Killers) NK and cytotoxic CD8$^+$ T cells using the granzyme/perforin mechanism [62]. Today, it is evident that this mechanism is not just restricted to NK and TCD8+, and T$_{regs}$ also employ this to induce cell cytolysis in TCD8$^+$, NK, and B effector cells in TME [63–65]. According to the study of Grossman et al., CD4$^+$CD25$^+$ natural T$_{regs}$ express more GZMA and less *GZMB*, whereas adaptive T$_{regs}$ express these molecules differently [66]. Additionally, they demonstrated that both T$_{reg}$ subtypes had a strong capacity for killing DC, T cells, and CD14$^+$ monocytes to suppress immune responses [66]. Our results also indicated that *GZMA* expression levels in T$_{regs}$ had increased following treatment, which may be a sign of their increased capacity for cell cytolysis of TME cells.

*IL-32* is a pro-inflammatory cytokine produced by different immune cells, including NK cells, T cells, and monocytes [67]. This cytokine has been shown to play a role in the induction of other pro-inflammatory cytokines like *IL-6*, *IL-8*, *TNF-α*, and macrophage inflammatory protein-2 (MIP-2) [67]. The decrease in the expression of *IL-32* in T$_{regs}$ after treatment can be an inhibitory mechanism of this cell to prevent the formation of inflammation in TME.

The importance of immunosuppressive TME and dysfunctional expression of IICs in reducing anti-tumoral immune responses has been underlined by increasing evidence [11]. Based on our findings, five of the IICs had significant expression changes when the IIC panel in T$_{regs}$ was compared before and after treatment (Fig 5C). According to these findings, PD-1$^+$ T$_{regs}$ significantly increased *LAG-3* expression after Nivolumab treatment, while *TIGIT*, *ENTPD1*, *CD276*, and *LY9* expression decreased significantly. *LAG-3* is a CD4-dependent molecule [68]. This molecule is produced on the surface of T$_{regs}$ after activation, binds to MHC-II, and inhibits DCs' and effector T cells' functions [68]. In addition, it has been demonstrated that LAG-3$^+$ T$_{regs}$ can control humoral immune responses [69]. The production of high amounts of *IL-10* and *TGF-β* by LAG-3$^+$ T$_{regs}$ inhibits the function of T$_{fh}$, prevents the formation of the germinal center (GC), and finally suppresses the production of antibodies [70]. The increase of LAG-3 expression in T$_{reg}$ after Nivolumab injection can increase their ability to suppress cellular and humoral immune responses. It should be emphasized that after the treatment, *CTLA-4* and *PDCD-1* expression levels also increased; however, this increase was not significant.

TIGIT is one of the most important IICs on the surface of T$_{regs}$, capable of suppressing effector T cells, decreasing NK cytolysis activity, and inhibiting antibody formation [71]. Preclinical research suggests that blocking the *PD-1/PD-L1* pathway and *TIGIT* simultaneously promotes tumor rejection and has promise for patients with solid tumors [71]. Our findings demonstrate that anti-PD-1 (Nivolumab) therapy alone also reduces *TIGIT* expression in T$_{regs}$.

Understanding T$_{regs}$ immunosuppressive actions is an appealing therapeutic option for promoting anti-tumor immune responses. However, the distinct cell-cell interaction patterns, notify for further investigation. In this case, comparing the cell-cell contact before and after Nivolumab administration showed a reduction in T$_{reg}$ communication with most clusters after treatment (Fig 6A). Additionally, the most significant ligand-receptor pairs between T$_{regs}$ and TAMs were marked in Fig 5C. After receiving Nivolumab, the interaction of five pairs of receptors and ligands (*SPP1/CD44*, *HLA-E/KLRC2*, *HLA-E/KLRK1*, *ANXA1/FPR3*, and *CXCL9/FCGR2A*) significantly decreased (Fig 5C). According to Cheng et al., the *CD44-SSP1* axis is a control mechanism that prevents the proliferation of effector T cells in the TME [72]. Additionally, they demonstrated that once T$_{regs}$ were presented in the TME, the level of CD44 expression in those cells was reduced, increasing the ability of T$_{regs}$ to proliferate [72]. Additionally, they demonstrated that once T$_{regs}$ were presented in the TME, the level of *CD44* expression in those cells was reduced, and their proliferation ability increased [72]. Our study

revealed that Nivolumab therapy reduces *CD44-SSP1* interaction, which may have increased T$_{reg}$ proliferation potential.

*KLRC2* (*NKG2C*) and *KLRK1* (*NKG2D*) are activating receptors mainly expressed on the surface of NK cells and T cells [73]. By binding these receptors to their ligands, these cells become more capable of cytotoxicity [73]. Though Protein Atlas has confirmed a small expression of *KLRK1* on the surface of non-classical monocytes, we did not identify any studies that specifically addressed these two receptors on the surface of TAMs. Nevertheless, our findings support a decreased *NKG2D*/C-HLA-E interaction between TAMs and T$_{regs}$ after treatment. Yang et al. showed that *NKG2D*$^+$*CD4*$^+$ T cells kill T$_{regs}$ following the binding of *NKG2D* to its ligand [74]. Although *NKG2D*/C-dependent cytolysis in macrophages has yet to be confirmed, we do know that these cells can directly kill other cells after stimulation by releasing lethal mediators such as TNF-α, reactive nitrogen species, and reactive oxygen species (ROS) [75]. The reduction of *NKG2D*/C-HLA-E interaction after Nivolumab injection could be the mechanism by which T$_{regs}$ escape death, although this data needs further study. Since the function of *FPR3* has not yet been determined [76], more experiments are essential to justify the effect of the reduction of the *FPR3-ANXA1* axis on T$_{reg}$ behavior after treatment with Nivolumab.

Furthermore, the reduction in T$_{reg}$-TAM interaction following Nivolumab therapy has influenced several mechanisms, including the inflammatory response, apoptosis, *IL-6/JAK/STAT3* signaling, *IL2/STAT5* signaling, and *TNF-α* signaling via NF-κB, though our data do not show the nature of these changes.

Our results showed that the increase in the number of T$_{regs}$ after Nivolumab treatment could be due to the reduction of the *CD44-SSP1* axis, KRAS signaling pathway, and *NKG2C*/D-HLA-E axis, which respectively lead to increased T$_{regs}$ proliferation, *FOXP3* expression, T$_{regs}$ differentiation, and the prevention of destruction by other cells (Fig 1).

The decrease in KRAS and Notch signaling pathway activity, which increases the expression of *FOXP3*, *PD-1*, and *IL-35*, can also contribute to the increased suppressive capacity of T$_{regs}$. These cells exhibit elevated *CTLA-4* and *LAG-3* expression on their surface in addition to *PD-1* and *TIGIT* expression, which inhibit the activity of T effectors and dendritic cells (DCs), prevent the development of GC, and inhibit the production of antibodies. Cellular and humoral immune systems can be suppressed more effectively by these cells. T$_{regs}$ produce more anti-inflammatory cytokines such as *IL-10*, *TGF-β*, and *IL-35* while producing less pro-inflammatory cytokine *IL-32*. The potential of these cells to destroy other cells has increased along with *GZMA* synthesis. Finally, different biological processes, including those related to neutrophil degranulation and activation, changed in the T$_{regs}$ of HNSCC patient's post-Nivolumab therapy. Although it is unknown from our data what kind of modifications these are (Fig 1).

Although reducing the immunosuppressive mechanisms and restoring the antitumor power of immune cells in the TME are of the IIC therapy goals, our study provided completely different results. T$_{regs}$, the most effective cells at inducing TME, are regarded as one of the most important targets of IIC therapy. However, our results did not only not show a decrease in the number and suppressive function of T$_{regs}$, but all the evidence indicated an increase in the number and intensification of the suppressive function of T$_{regs}$.

Finally, by studying other articles, the probability of hyper progressive disease (HPD) in these four HNSCC patients after being treated with Nivolumab was strengthened. A subset of cancer patients receiving CPI experience enhanced tumor cell proliferation, rapid disease progression, and a poor prognosis [77–84]. This condition known as HPD is seen in 4–29% of cancer patients [85]. Here are still no specific criteria to predict the probability of HPD occurrence in patients. In addition, the mechanisms involved in HPD remain unclear. However, some potential HPD mechanisms have been proposed, including T$_{regs}$ proliferation, up-regulation of *CTLA-4* and *LAG-3*, an increase in ILC-3, etc. [48]. Further studies of these

mechanisms can probably help clarify the limitations of IIC therapy success in HNSCC patients and identify effective therapeutic targets for HPD patients.

## Conclusion

In conclusion, this study investigated the dynamic changes in the TME of HNSCC patients treated with Nivolumab, with a particular focus on $T_{regs}$. DEG analysis of $T_{regs}$ revealed down-regulation of genes associated with immunosuppressive functions, and upregulation of immediate early genes. Biological process and MSigDB analysis showed negative regulation of KRAS signaling, Notch signaling, and angiogenesis in $T_{regs}$ after Nivolumab therapy. Additionally, the study found altered expression of certain IICs in $T_{regs}$ before and after treatment. These findings provide new insights into the response to Nivolumab therapy in HNSCC patients, and suggest that targeting $T_{regs}$ and IICs could potentially enhance the efficacy of immunotherapy.

## Supporting information

**S1 Fig. PCA and UMAP of cell in both before and after batch effect correction.** (JPG)

**S1 File.** (PDF)

## Author Contributions

**Conceptualization:** Adib Miraki Feriz, Fatemeh Bahraini, Arezou Khosrojerdi, Setareh Azarkar.

**Data curation:** Seyed Mehdi Sajjadi, Edris HosseiniGol.

**Formal analysis:** Adib Miraki Feriz, Fatemeh Bahraini.

**Investigation:** Arezou Khosrojerdi, Setareh Azarkar, Mohammad Amin Honardoost, Samira Saghafi.

**Methodology:** Adib Miraki Feriz.

**Project administration:** Hossein Safarpour, Vito Racanelli.

**Software:** Adib Miraki Feriz, Edris HosseiniGol.

**Supervision:** Hossein Safarpour, Vito Racanelli.

**Validation:** Adib Miraki Feriz, Hossein Safarpour.

**Visualization:** Setareh Azarkar, Mohammad Amin Honardoost, Nicola Silvestris.

**Writing – original draft:** Adib Miraki Feriz, Fatemeh Bahraini, Arezou Khosrojerdi, Setareh Azarkar, Seyed Mehdi Sajjadi, Patrizia Leone.

**Writing – review & editing:** Arezou Khosrojerdi, Seyed Mehdi Sajjadi, Samira Saghafi, Nicola Silvestris.

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
