## [Decision Letter · Decision Letter 0]

23 Aug 2023

PONE-D-23-21092Deciphering the Immune Landscape of Head and Neck Squamous Cell Carcinoma: A Single-Cell Transcriptomic Analysis of Regulatory T Cell Responses to PD-1 Blockade TherapyPLOS ONE

Dear Dr. Safarpour,

Thank you for submitting your manuscript to PLOS ONE. After careful consideration, we feel that it has merit but does not fully meet PLOS ONE’s publication criteria as it currently stands. Therefore, we invite you to submit a revised version of the manuscript that addresses the points raised during the review process.

We look forward to receiving your revised manuscript.

Kind regards,

Lu Zhang

Academic Editor

PLOS ONE

2. Please provide Table S1 mentioned in line 356.

5. We note that Figure 1 in your submission contain copyrighted images. All PLOS content is published under the Creative Commons Attribution License (CC BY 4.0), which means that the manuscript, images, and Supporting Information files will be freely available online, and any third party is permitted to access, download, copy, distribute, and use these materials in any way, even commercially, with proper attribution. For more information, see our copyright guidelines: http://journals.plos.org/plosone/s/licenses-and-copyright.

Reviewers' comments:

Reviewer's Responses to Questions

**Comments to the Author**

1. Is the manuscript technically sound, and do the data support the conclusions?

Reviewer #1: Partly

Reviewer #2: Yes

2. Has the statistical analysis been performed appropriately and rigorously? 

Reviewer #1: I Don't Know

Reviewer #2: Yes

3. Have the authors made all data underlying the findings in their manuscript fully available?

Reviewer #1: No

Reviewer #2: Yes

4. Is the manuscript presented in an intelligible fashion and written in standard English?

Reviewer #1: Yes

Reviewer #2: Yes

5. Review Comments to the Author

Reviewer #1: This paper presents a study to investigate the effect of Nivolumab on the function and cellular communication of Tregs using scRNA-seq data analysis. Authors gained a better understanding of the mechanisms underlying the response and resistance to anti-PD-1 CPI in HNSCC. This study had the potential to contribute to the development of more effective immunotherapeutic strategies for the treatment of HNSCC. However, there are questions that limit my enthusiasm of the paper, as outlined below.

1- How can authors generalize the findings based on using only one scRNA-seq data (GSE195832)? Any validation study to assess the performance of findings.

2- In data section, add more details about the TCGA data and normalized expression data (e.g., TPM).

3- Authors exp-lain the PCA and combat analyses to remove potential variation or batches. Please add the plots and more details as supplementary file. More visualization plots related to before and after correction (using top 20 PCS!!). How are the technical variations associated with age and sex?

4- For the scRNAseq data, authors only mentioned normalization approach, while need more details about pre-processing steps.

5- Enrichment analysis using two different web-apps. The Enrichr web-app is based on ORA while the other one was mentioned on line 390 also included the ORA. Please clarify this part. Be more precise about where or for which analysis the GSEA or ORA were applied.

6- Authors mentioned different web-app, EnrichR, GEPIA database, SquidPy, scFates, CellPhone DB, Pseudo-time trajectory analysis, etc., while not enough details were shared about these web-apps including methods briefly, etc. Not easy to follow to readers.

7- Which software was used? Any codes to be shared through GitHub?

8- How the canonical gene markers were selected (CD3D, FOXP3, TIGIT, and FANK1)?

9- Figure 2D add the y-axis label. What is the statistical method to compare two groups (N vs T)? Figure 2G needs to re-generate. There IS NOT Figure 2H.

10- DEG analysis: There are some questions about

a. Is the t-test paired or two group independent?

b. Fig 3A does not make sense. Why there are some outliers (log FC > 10 or < -10)? Plot is confusing any cut-off for log FC? It is not clear on figure the legend. Why authors chose the DEGs based on p-value and not the FDR or corrected p-value?

11- Fig 3B is based on ORA analyses? Add at the caption. Fig 3D, what is the y-axis? Why the G1 has not defined on caption. More precise about p-value or FDR (corrected p-value). The DEG or pathway results should be based on FDR.

12- Fig 4A, what do the rows and columns represent? why not clustering for genes? the color bar is missing. How the top 10 genes (up-regulating selected). It is not clear from heatmap.

13- Fig 4c, I assume the y-axis represent the log-expression transformed. If yes, why here is -1 or less than zero?

14- Figure 5, same comments as Figures 2 and 4.

15- How was the cell-cell interaction assessed? Any metrics or visualization plot to show how the 5 pairs were selected? Did you measure the correlation across ligand-receptor pairs? The enrichment analysis (Fig 6D) was applied based on the 10 genes? Please add more details about the enrichment analysis.

16- All the abbreviations should be defined in advance (e.g., TAM).

Reviewer #2: The authors have re-analysed the published scRNAseq data. The manuscript submitted is technically sound and well written. Statical analysis are appropriately done and the given data support their conclusions. However, if accepted for publication, the clarity of the figures should be improved (high resolution figures). The figure markings are not visible properly in the present form.

6. PLOS authors have the option to publish the peer review history of their article (what does this mean?). If published, this will include your full peer review and any attached files.

Reviewer #1: No

Reviewer #2: No

---

## [Author Response · Author response to Decision Letter 0]

5 Nov 2023

We thank the two anonymous reviewers for taking the time to carefully assess our manuscript and provide valuable comments and suggestions. We appreciate the opportunity to provide this revised manuscript for further consideration.

---

## [Editor Report · Decision Letter 1]

1 Dec 2023

Deciphering the Immune Landscape of Head and Neck Squamous Cell Carcinoma: A Single-Cell Transcriptomic Analysis of Regulatory T Cell Responses to PD-1 Blockade Therapy

PONE-D-23-21092R1

Dear Dr. Safarpour,

We’re pleased to inform you that your manuscript has been judged scientifically suitable for publication and will be formally accepted for publication once it meets all outstanding technical requirements.

Kind regards,

Lu Zhang

Academic Editor

PLOS ONE
---

## [Editor Report · Acceptance letter]

6 Dec 2023

PONE-D-23-21092R1 

Deciphering the Immune Landscape of Head and Neck Squamous Cell Carcinoma: A Single-Cell Transcriptomic Analysis of Regulatory T Cell Responses to PD-1 Blockade Therapy 

Dear Dr. Safarpour:

I'm pleased to inform you that your manuscript has been deemed suitable for publication in PLOS ONE. Congratulations! Your manuscript is now with our production department. 

Kind regards, 

on behalf of

Dr. Lu Zhang 

Academic Editor

PLOS ONE